# The homologous recombination machinery modulates the formation of RNA–DNA hybrids and associated chromosome instability

Lamia Wahba[1,2], Steven K Gore[1†], Douglas Koshland[1]*

[1]Department of Cell and Molecular Biology, Howard Hughes Medical Institute, University of California, Berkeley, Berkeley, United States; [2]Department of Biology, Johns Hopkins University, Baltimore, United States

**Abstract** Genome instability in yeast and mammals is caused by RNA–DNA hybrids that form as a result of defects in different aspects of RNA biogenesis. We report that in yeast mutants defective for transcription repression and RNA degradation, hybrid formation requires Rad51p and Rad52p. These proteins normally promote DNA–DNA strand exchange in homologous recombination. We suggest they also directly promote the DNA–RNA strand exchange necessary for hybrid formation since we observed accumulation of Rad51p at a model hybrid-forming locus. Furthermore, we provide evidence that Rad51p mediates hybridization of transcripts to homologous chromosomal loci distinct from their site of synthesis. This hybrid formation in *trans* amplifies the genome-destabilizing potential of RNA and broadens the exclusive co-transcriptional models that pervade the field. The deleterious hybrid-forming activity of Rad51p is counteracted by Srs2p, a known Rad51p antagonist. Thus Srs2p serves as a novel anti-hybrid mechanism in vivo.

*For correspondence: koshland@berkeley.edu

†Present address: University of California, Davis, Davis, United States

Competing interests: The authors declare that no competing interests exist.

## Introduction

Genome instability can lead to a range of alterations in both the sequence and structure of chromosomes. While such changes may help drive evolution, more often they are associated with decreased organism fitness and increased susceptibility to disease (*Aguilera and Gómez-González, 2008*). Historically, most genome instability was thought to occur as the result of errors during replication, or the failure of DNA repair pathways. However, work in *Saccharomyces cerevisiae* and mammalian cells has demonstrated that genome instability also arises from lesions generated from the formation of RNA–DNA hybrids (*Huertas and Aguilera, 2003*; *Li and Manley, 2005*; *Kim and Jinks-Robertson, 2009*; *Paulsen et al., 2009*; *Wahba et al., 2011*; *Stirling et al., 2012*). Many important aspects of hybrid-mediated genome instability remain to be elucidated.

Genome-wide screens in budding yeast and human cells have revealed that levels of RNA–DNA hybrids increase when RNA biogenesis is disturbed at sites of transcription initiation or repression, elongation, splicing, degradation, and export (*Huertas and Aguilera, 2003*; *Li and Manley, 2005*; *Paulsen et al., 2009*; *Wahba et al., 2011*; *Stirling et al., 2012*). The co-transcriptional binding of many RNA processing and transcription factors suggests that they prevent hybrid formation by restricting the access of nascent RNA molecules to the DNA template at the site of transcription (*Aguilera and García-Muse, 2012*). Recent studies suggest that these RNA biogenesis factors are not sufficient to prevent transient hybrid formation at some loci in wild-type budding yeast; rather, hybrids form but are removed rapidly by hybrid removal factors, including two endogenous RNase H enzymes, and Sen1p, an RNA–DNA helicase (*Mischo et al., 2011*; *Wahba et al., 2011*). In RNA biogenesis mutants

**eLife digest** Cells with an unusually large number of mutations—either in the form of changes to the DNA sequence or changes in the number or structure of chromosomes—are said to show genome instability. Although these mutations sometimes boost an organism's chances of survival and reproduction, they more often have detrimental effects, which can include cancer.

Genome instability can arise as a result of mistakes occurring during the repair of damaged DNA, or due to inappropriate hybridization of RNA to its DNA template. These RNA–DNA hybrids had been thought to occur strictly during the transcription of DNA into RNA. During this process, the two strands of the DNA molecule separate behind the moving RNA polymerase, and this provides an opportunity for the newly formed RNA to hybridize back to its DNA template. When these RNA–DNA hybrids persist, they give rise to DNA damage that leads to genome instability.

Although much is known about the factors that prevent the formation of hybrids, or promote their removal, little is known about how hybrids form in the first place. Now, Wahba et al. have identified one such mechanism in the model yeast, *Saccharomyces cerevisiae*. It involves a protein called Rad51p, which helps to join stretches of nucleic acids together to repair breaks in DNA. However, Wahba et al. showed that if Rad51p is not properly regulated, it can also trigger the formation of RNA–DNA hybrids; yeast cells that lack the gene for Rad51p showed significantly reduced levels of hybrid formation. Moreover, dysfunctional Rad51p causes RNA sequences to anneal to DNA throughout the genome, rather than just at the site in which the RNA was originally produced. This means that RNA sequences produced during transcription are much more of a threat to genomic stability than previously thought.

The work of Wahba et al. presents a paradox in which a protein that is normally involved in repairing DNA can itself cause damage if it is not carefully regulated. It also raises the possibility that the elevated levels of Rad51p expression observed in cancer cells could be a cause, rather than a consequence, of mutations.

the elevated levels of hybrid formation overwhelm the capacity of these hybrid removal factors, allowing the accumulation of hybrids and genome instability.

While a number of factors that prevent hybrid formation or persistence have been identified, little is known about factors that promote the formation of RNA–DNA hybrids in vivo. One potential factor is RNA polymerase, which generates negative supercoiling behind the elongating polymerase. The negative supercoiling facilitates DNA unwinding, and may allow RNA access to the DNA template (*Roy et al., 2010*). A connection between negative supercoiling and hybrid formation is supported by in vivo work on topoisomerase mutants in bacteria, yeast, and human cells (*Drolet et al., 1995*; *Tuduri et al., 2009*; *El Hage et al., 2010*). Another potential promoter of hybrid formation is RecA, the bacterial strand exchange protein that normally promotes the invasion of single-stranded DNA into duplex DNA to repair DNA damage. Studies a decade ago showed that RecA promotes RNA–DNA hybrid formation in vitro (*Kasahara et al., 2000*; *Zaitsev and Kowalczykowski, 2000*). This observation supported a model of RecA-dependent hybrid formation that had been postulated as an alternative mechanism to initiate DNA replication (*Cao and Kogoma, 1993*; *Hong et al., 1995*). The intriguing possibility that RecA or its eukaryotic ortholog, Rad51p, might play a role in vivo to promote hybrid formation has not been pursued further.

The in vitro studies on RecA-dependent hybrid formation also challenged ideas of when hybrid formation occurs. In these studies, RecA-mediated R-loops formed when RecA was mixed with RNA and its homologous DNA template in the absence of active transcription. This observation showed that hybrids could, at least in vitro, form post-transcriptionally (i.e., in *trans*) (*Kasahara et al., 2000*; *Zaitsev and Kowalczykowski, 2000*). However, to date, models postulated from in vivo studies suggest that RNA–DNA hybrids occur co-transcriptionally (in *cis*) from the invasion of the duplex DNA at the site of transcription by a nascent RNA transcript (*Huertas and Aguilera, 2003*; *Aguilera and García-Muse, 2012*). The *cis* mechanism is supported by the co-transcriptional nature of many of the RNA biogenesis steps implicated in preventing hybrid formation. However, the *cis* mechanism does not fully explain how mutants in post-transcriptional processes, such as RNA degradation and export, would cause hybrid-mediated instability (*Wahba et al., 2011*). Therefore, the formation of RNA–DNA

hybrids may occur in *trans* as well as in *cis*. The intriguing possibility that hybrids may occur in *trans* and contribute to genomic instability has not been assessed.

In this work we used *S. cerevisiae* as the model system to test in vivo the role of Rad51p in hybrid formation. We report that the formation of RNA–DNA hybrids and associated genome instability in at least four RNA biogenesis mutants requires Rad51p and its activator, Rad52p. Furthermore, the deleterious hybrid-forming activity of Rad51p is suppressed in wild-type cells by Srs2p, a Rad51p inhibitor. Additionally, we developed a model locus system that allows us to monitor hybrid-mediated genome instability as a result of transcription. We manipulate this system to provide compelling evidence that hybrids and ensuing genome instability can occur via a *trans* mechanism that is dependent on Rad51p.

## Results

### Formation of RNA–DNA hybrids is dependent on Rad51p

The conditions that drive the initial formation of RNA–DNA hybrids in vivo are not well understood. With the bacterial in vitro experiments in mind, we wondered whether hybrid formation was simply a strand exchange reaction, similar to that mediated by Rad51p during DNA repair and homologous recombination. To test this possibility, we examined the effect of deleting *RAD51* on hybrid formation and the associated genome instability in RNA biogenesis mutants of budding yeast. We chose a representative set of mutants defective in elongation (*leo1Δ*), repression (*med12Δ* and *sin3Δ*), and degradation (*kem1Δ* and *rrp6Δ*). We assayed directly for the presence of RNA–DNA hybrids in wild-type cells and these mutants by staining chromosomes in spread nuclei with S9.6 antibody (see 'Materials and methods'). Previously, we demonstrated the specificity of the S9.6 antibody for hybrids by two approaches. First, S9.6 staining in spreads of RNA biogenesis mutants is reduced to that seen in wild-type cells by post treatment of chromosome spreads with RNase H (*Wahba et al., 2011*). Similarly, spreads of an RNA biogenesis mutant over-expressing RNase H no longer stained with S9.6.

As reported previously, less than 5% of wild-type nuclei stain with this antibody (*Figure 1A*, *Figure 1—figure supplement 1*). In contrast, 80–85% of nuclei in our representative set of RNA biogenesis mutants showed robust staining, indicating the formation of stable hybrids at many loci in most cells (*Figure 1A*, *Figure 1—figure supplement 1*). The deletion of *RAD51 (rad51Δ)* in these mutants diminished S9.6 staining in nearly all nuclei from the RNA biogenesis mutants threefold to fourfold to near background levels (*Figure 1A*, *Figure 1—figure supplement 2*). To corroborate our cytological method, we isolated total nucleic acids from wild-type, *sin3Δ* (a representative RNA biogenesis mutant), and *sin3Δ rad51Δ* cells, transferred them to a solid matrix and monitored binding of S9.6. S9.6 binding to *sin3Δ* nucleic acids was elevated approximately tenfold relative to *sin3Δ rad51Δ* (*Figure 1—figure supplement 3*). These results strongly suggest that hybrid formation in these mutants is highly dependent upon Rad51p.

One prediction from the cytological results is that the suppression of hybrid formation by *rad51Δ* should also lead to the suppression of hybrid-mediated chromosome instability. To measure hybrid-mediated genome instability, we exploited an assay we developed previously using a yeast artificial chromosome (YAC) (*Wahba et al., 2011*; see 'Materials and methods'). The total rate of YAC instability (the sum of chromosome loss and terminal deletions) in wild-type cells was $6 \times 10^{-4}$ per division. Notably, *rad51Δ* alone caused no increase in YAC instability. In our subset of RNA biogenesis mutants, YAC instability increased fivefold to tenfold (*Figure 1B*). The introduction of *rad51Δ* into the RNA biogenesis mutants completely suppressed the elevated YAC instability, both chromosome loss and terminal deletions, in *leo1Δ*, *kem1Δ*, *rrp6Δ*, and *sin3Δ* mutants. In *med12Δ*, YAC instability was mostly but not entirely suppressed despite the near complete suppression of hybrid formation as monitored by spreads, indicating that in the *med12Δ rad51Δ* strain a subset of the YAC instability was hybrid independent. Overall, the suppression of hybrid-mediated chromosome instability by *rad51Δ* corroborates its elimination of RNA–DNA hybrids and associated destabilizing lesions.

To further validate the occurrence of Rad51-dependent hybrids, we sought to develop a model locus that can be used to induce hybrid formation and hybrid-mediated instability at a known region. From our previous study on RNA biogenesis mutants that induce hybrids, we noted that many of these mutants allow cryptic transcription, and likely the production of aberrant transcripts (*Wyers et al., 2005*; *Cheung et al., 2008*; *Wahba et al., 2011*). Based on this observation, we introduced a portion of the GAL1-10 promoter into the YAC (henceforth referred to as YAC-GALpr), such that the addition of galactose to the media would induce GALpr-dependent transcription of neighboring non-yeast

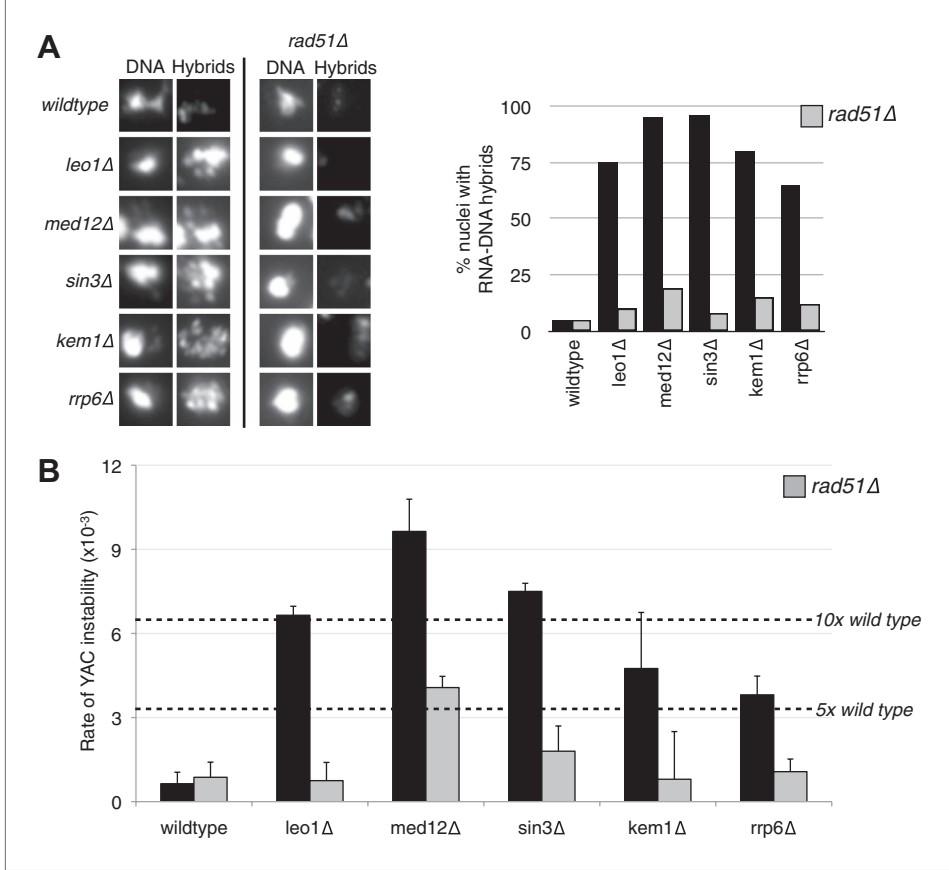

**Figure 1**. Deletion of RAD51 suppresses RNA–DNA hybrids and YAC instability. (**A**) Left panel: Representative images of chromatin spreads stained with S9.6 antibody, showing reduced RNA–DNA hybrid staining in mutants with a deletion of *RAD51* (*rad51Δ*). Right panel: The percent of total nuclei scored that stain positively for RNA–DNA hybrid in chromatin spreads is quantified. A total of 50–100 nuclei from two independent experiments were scored for each genotype. (**B**) Rate of yeast artificial chromosome (YAC) instability in mutants is also reduced when *RAD51* is deleted. Error bars represent standard deviation calculated from at least four independent colonies.

The following figure supplements are available for figure 1:

**Figure supplement 1**. Larger panels of chromatin spreads showing multiple nuclei of single mutants stained with S9.6 antibody.

**Figure supplement 2**. Larger panels of chromatin spreads showing multiple nuclei of double mutants stained with S9.6 antibody.

**Figure supplement 3**. Dot blotting with S9.6 antibody.

sequences (*Figure 2A*). We analyzed transcription of the human and vector sequences flanking GALpr by qRT-PCR. This analysis revealed approximately one hundredfold induction of RNA at least 1 kb on both sides of the GAL promoter (*Figure 2B*).

Using the model locus, we monitored the presence of transcription-induced hybrids specifically proximal to GALpr. Total nucleic acids were isolated from strains containing either the YAC or YAC-GALpr in the presence or absence of galactose. These samples were subjected to DNA immunoprecipitation (DIP) analysis with the S9.6 antibody that should only precipitate DNA in RNA–DNA hybrids (*Mischo et al., 2011*; see 'Materials and methods'). Using primers specific to the YAC region proximal to the GALpr, low DIP signals were observed in YAC-GALpr cultures in the absence of galactose, as well as in cultures with the YAC, with and without the addition of galactose (*Figure 2C*, *Figure 2—figure supplement 1A*). Thus, hybrids form rarely in the YAC sequences proximal to the GALpr in the

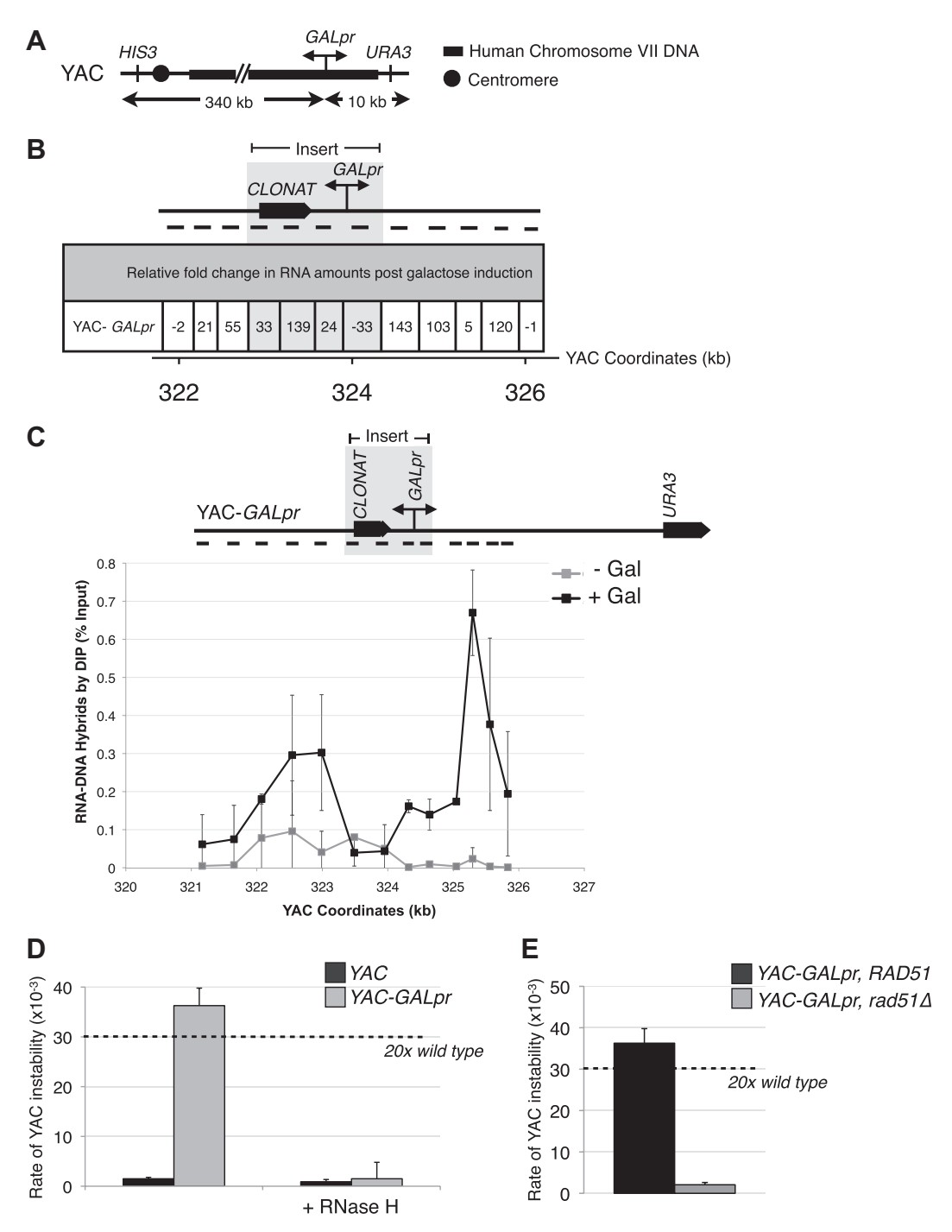

**Figure 2**. Hybrid-mediated YAC instability is induced in wild-type when high rates of transcription are induced on the YAC using the GAL1-10 promoter (*GALpr*). (**A**) Schematic of the YAC-GALpr construct. Total yeast artificial chromosome (YAC) length is 350 kb, of which 324 kb come from human chromosome VII. The GALpr was integrated 10 kb from the telomere, on the arm with the *URA3* marker. (**B**) Quantitative RT-PCR monitoring changes in RNA levels on the YAC 5 hr after induction with galactose. YAC RNA is normalized to actin RNA, and represented as fold change, as compared to RNA levels detected in uninduced cells. Above the table is a schematic representation of the YAC region from which RNA is measured, with the qRT-PCR fragments used in quantification indicated with black dashes. The region in gray represents the GAL1-10 promoter and selectable marker integrated in the YAC-GALpr strain. (**C**) DIP analysis to monitor RNA–DNA hybrid formation in the YAC-GALpr strain in the absence of galactose, and 2 hr after induction with galactose. Error bars represent standard deviation calculated from two independent DIP experiments. (**D**) Rates of YAC instability in strains with *YAC* (black bars) or YAC-GALpr (gray bars) 5 hr after addition of galactose to the media. Strains carried either an RNase H over-expressing

*Figure 2. Continued on next page*

*Figure 2. Continued*
plasmid or an empty control vector. (**E**) Induced YAC instability is suppressed when *RAD51* is deleted. Error bars represent standard deviation calculated from at least three independent colonies.
The following figure supplements are available for figure 2:

**Figure supplement 1**. (A) DIP analysis of YAC strain prior to and 2 hr after addition of galactose to the media. (B) Monitoring of DIP signal in the YAC-GALpr strain at a distal region, showing low levels of hybrid signal upon induction with galactose as compared to. (C) DIP signals are reduced around the YAC-GALpr module upon return to repressive conditions.

**Figure supplement 2**. The percent of terminal deletions and chromosome loss events recovered after 5 hr of growth in galactose-containing media is comparable for YAC and YAC-GALpr strains.

absence of their transcription. In contrast, we observed a dramatic increase in the DIP signal for hybrids on the YAC sequences proximal to YAC-GALpr, 2 hr after induction by galactose (*Figure 2C*). The specificity of this increased DIP signal was evident by the fact that no elevation in hybrid signal was detected in two regions of the YAC-GALpr distal to the GALpr (*Figure 2—figure supplement 1B*). Additionally, lower DIP signals coincide with the transcriptional start site of the GALpr, where there is little transcript detectable upon addition of galactose (*Figure 2C*). Furthermore, the DIP signal in the YAC-GALpr strains was suppressed when transcription was repressed by the addition of glucose (*Figure 2—figure supplement 1C*). Finally, hybrid formation at YAC-GALpr was dependent upon RAD51 (see 'Rad51p-dependent hybrid formation can occur in trans'). These data provide molecular evidence for the formation of RAD51-dependent hybrids at the YAC sequences transcribed by induction of GALpr.

To determine whether the Rad51p-dependent hybrids induced by YAC-GALpr led to genome instability, we monitored the instability of YAC-GALpr upon galactose treatment. Indeed, its instability was elevated 25-fold with a distribution of chromosome loss and terminal deletions similar to that seen in wild-type cells and RNA biogenesis mutants (*Figure 2*, *Figure 2—figure supplement 2*). Furthermore, this transcription-induced YAC instability was suppressed by over-expression of RNase H or deletion of *RAD51* (*Figure 2D–E*). Thus both by DIP and YAC instability, hybrids induced by transcription at the model YAC-GALpr locus, like those induced by RNA biogenesis mutants, required Rad51p for their formation.

## Rad51p binding at the site of hybrid formation

A second prediction concerning Rad51p-mediated hybrid formation is that Rad51p should be detectable near sites of hybrid formation. To test this prediction we used our YAC-GALpr model locus to assay for the presence of Rad51p binding around the site of hybrid formation. We generated cultures of strains containing the YAC or YAC-GALpr that had been grown in the presence or absence of galactose. These cultures were fixed and assayed for Rad51p binding to the YAC sequences by chromatin immunoprecipitation (ChIP) (see 'Materials and methods'). ChIP was performed using two independent antibodies, anti-HA against a C-terminal haemagglutinin (HA) tagged Rad51p and a polyclonal rabbit anti-Rad51p. No Rad51p binding was detected either on YAC-GALpr in the absence of galactose or on the YAC in the presence or absence of galactose (*Figure 3A*). Thus the level of Rad51p binding to the YAC or vector sequences in the absence of transcription was very low if any. In contrast, using either antibody for ChIP, significant Rad51p binding was detected around the GAL promoter on YAC-GALpr upon the addition of galactose and induction of transcription (*Figure 3A*, *Figure 3—figure supplement 1*). Notably, Rad51p binding appears to extend further than the region of hybrid formation detected by DIP (*Figure 2C* and *Figure 3A*). Rad51p is known to spread from regions of ssDNA into dsDNA (*Zaitsev and Kowalczykowski, 2000*), and it is possible that in our model locus Rad51p is spreading from the ssDNA or RNA–DNA hybrid into the neighboring dsDNA. To test further the correlation of transcription and Rad51p binding, we added dextrose to the galactose-treated YAC-GALpr cultures to repress galactose-induced YAC-GALpr transcription (see 'Materials and methods'). In these cultures, Rad51p binding disappeared (*Figure 3—figure supplement 2*). Taking these findings together, we observe Rad51p binding to the region of the hybrid-forming locus on the YAC-GALpr only when transcripts from this region are induced.

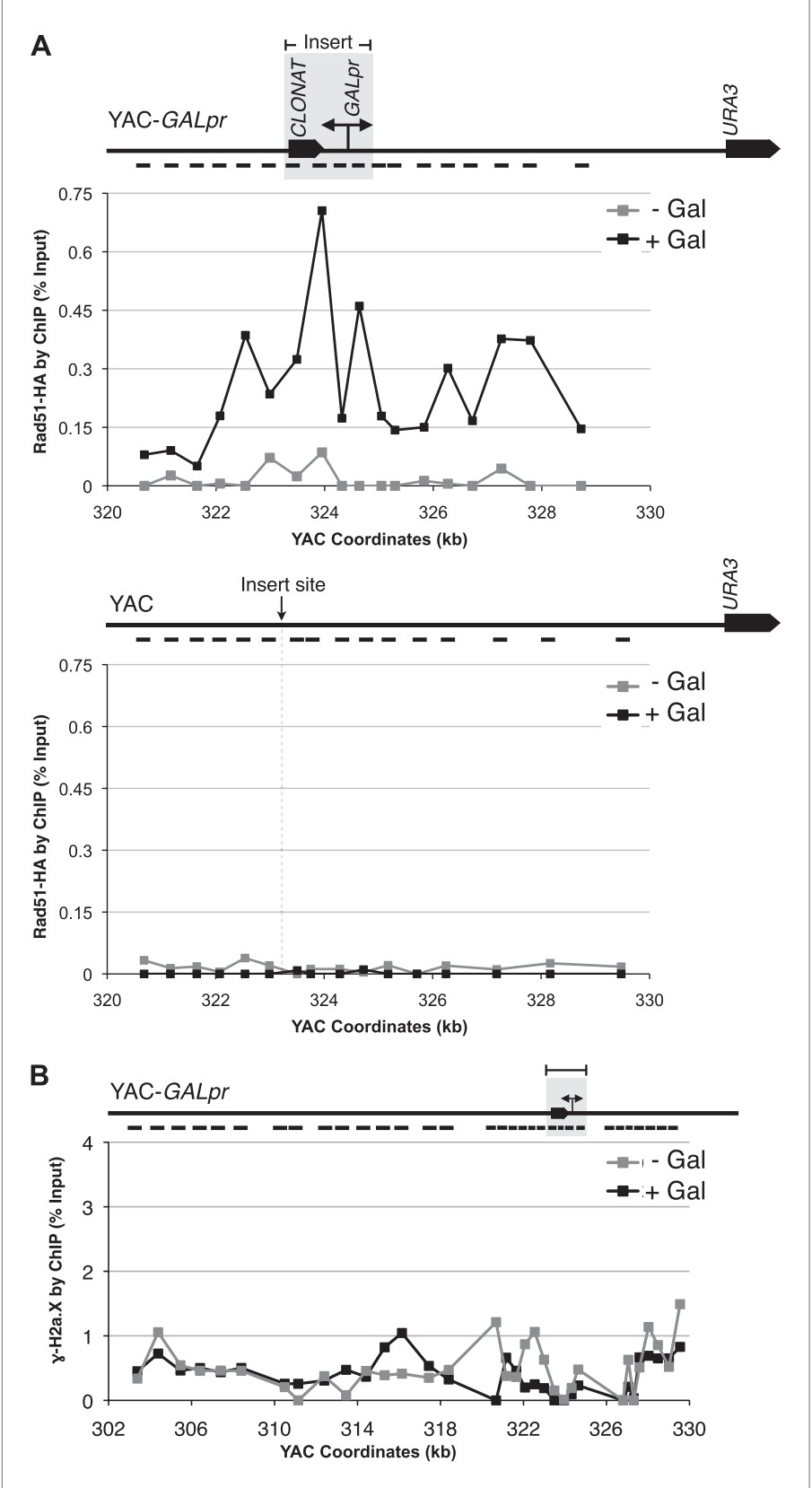

**Figure 3**. Rad51p binding is detectable around the YAC-GALpr module upon induction of transcription. Cells growing exponentially in YEP-lactic acid were split, and galactose added to one half. The other half was collected

*Figure 3. Continued on next page*

*Figure 3. Continued*

immediately for the –Gal sample and fixed for chromatin immunoprecipitation (ChIP; see 'Materials and methods'). After 120 min, the +Gal sample was similarly fixed for ChIP. Input DNA and DNA coimmunprecipitated with α-HA or -γ-H2a.X (IP) antibody were amplified using primer sets along the yeast artificial chromosome (YAC) as annotated with black dashes on the YAC-GALpr or YAC schematic above each graph. (**A**) ChIP of Rad51-HA in the YAC-GALpr strain shows an increased signal in Rad51-HA binding 2 hr after induction of transcription by addition of galactose to the media (top panel). No change in the RAD51-HA signal is observed in the YAC strain (bottom panel). (**B**) ChIP of γ-H2a.X in YAC-GALpr reveals no significant change in signal within 2 hr of galactose induction.

The following figure supplements are available for figure 3:

**Figure supplement 1**. Rad51p binding is detectable around the YAC-GALpr module upon induction of transcription.

**Figure supplement 2**. Rad51p binding is reduced around the YAC-GALpr module upon return to repressive conditions.

**Figure supplement 3**. Rad51 and γ-H2a.X binding at an inducible break site on chromosome III.

**Figure supplement 4**. Wild-type levels of YAC instability are observed after 2 hr of transcription induction.

We propose that the binding of Rad51p observed at the model locus is due to its role in hybrid formation. However, hybrids are thought to induce double-strand breaks (DSBs), and Rad51p binds at DSBs to initiate DNA repair through homologous recombination (*Sugawara et al., 2003*; *Figure 3—figure supplement 3A*). Therefore, the presence of Rad51p at the hybrid-forming locus might be due to its function in repair rather than in hybrid formation. To address this alternative explanation for Rad51p binding, we performed molecular and functional tests for the formation of DSBs 2 hr after the induction of transcription. As a molecular assay, we monitored a 20 kb region surrounding the GAL promoter for the accumulation of phosphorylated histone H2AX (γ-H2AX) by ChIP. This modification is one of the most dramatic and earliest markers of DSB formation, arising within minutes and spanning large regions of chromatin adjacent to the break (*Shroff et al., 2004*; *Figure 3—figure supplement 3B*). However, we did not detect a ChIP signal for γ-H2AX above background level in the YAC-GALpr strain even under conditions that induced Rad51p binding (*Figure 3B*). Thus by this molecular assay Rad51p binding occurs at the site of hybrid formation prior to hybrid-induced DNA damage.

As a functional test, we took advantage of the fact that adding dextrose after 2 hr suppressed transcription and Rad51p binding at the model hybrid locus. We reasoned that if Rad51p binding during the 2 hr prior to the addition of dextrose reflected Rad51p association with hybrid-induced DNA damage, then this damage would manifest as increased YAC instability. However, no increase in YAC instability was observed (*Figure 3—figure supplement 4*), indicating that binding of Rad51p to this locus during the first 2 hr was unlikely to result from DNA damage. Thus neither our molecular nor functional test supports the binding of Rad51p to the model locus prior to hybrid-induced DNA damage, pointing to a direct role of Rad51p in hybrid formation.

Does the formation of all hybrids require Rad51p? Studies from our laboratory and the Aguilera laboratory suggest that hybrids not only form in RNA biogenesis mutants but also transiently in wild-type cells (*Mischo et al., 2011*; *Wahba et al., 2011*). The latter fail to persist because of their rapid removal by RNases H and Sen1 (*Mischo et al., 2011*; *Wahba et al., 2011*). To test whether these naturally occurring hybrids are also dependent on Rad51p, we monitored hybrid staining and YAC instability in *rnh1Δrnh201Δ* in the absence of *RAD51*. Neither hybrid staining nor YAC instability was suppressed (*Figure 4A,B*), indicating that the transient hybrids in wild-type cells are not Rad51p dependent. Thus both Rad51p-dependent and -independent mechanisms for hybrid formation exist.

## Rad51p-dependent hybrid formation can occur in *trans*

In the in vitro bacterial studies, RecA promoted hybrid formation in the absence of active transcription, suggesting that RNA–DNA hybrids can form post-transcriptionally, or in *trans*. To test whether in vivo

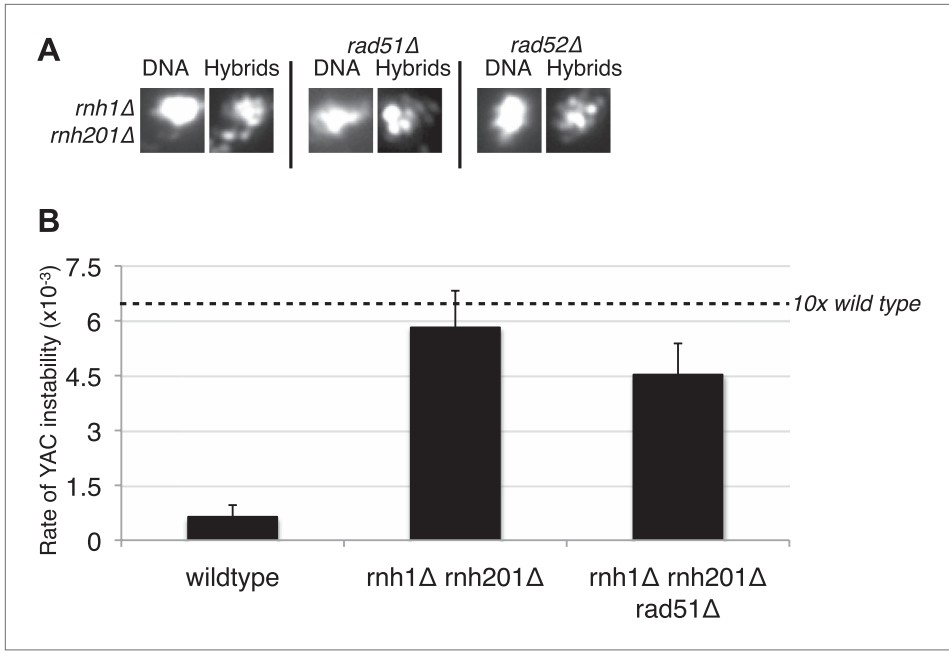

**Figure 4**. Deletions of RAD51 and RAD52 do not affect RNA–DNA hybrid formation in *rnh1Δrnh201Δ*.
(**A**) Representative images of chromatin spreads stained with S9.6 antibody. (**B**) Rate of yeast artificial chromosome
(YAC) instability is similar in *rnh1Δrnh201Δ* and strains lacking *RAD51* (*rad51Δ*) or *RAD52* (*rad52Δ*). Error bars
represent standard deviation calculated from at least six independent colonies.

hybrids could form in *trans*, we constructed a strain, LW7003, in which chromosome III contained
3.5 kb of vector and human sequences surrounding the galactose promoter of the YAC-GALpr (hence-
forth referred to as the YAC-GALpr module). This strain also contained the original unmodified YAC,
allowing us to investigate whether transcription of the YAC-GALpr module on chromosome III could
induce both hybrid formation on the YAC and YAC instability (*Figure 5A*).

To test directly whether hybrids can form in *trans*, DIP was performed on cultures of our LW7003
strain after growth in the presence or absence of galactose. One primer set that monitored hybrids
from both the YAC and YAC-GALpr module generated a strong DIP signal only in the presence of
galactose (*Figure 5B*, primer 1). This combined hybrid signal was eliminated when the *rad51Δ* was
introduced in this strain (*Figure 5B*, primer 1). These results minimally corroborate our previous dem-
onstration of hybrids forming in *cis* and show that hybrid formation is dependent upon Rad51p.

Two other primer sets that monitored hybrids only from the YAC also revealed a strong DIP signal
only in the presence of galactose (*Figure 5B*, primers 2 and 3). These results demonstrated transcription-
dependent hybrid formation in trans. This *trans*-specific hybrid signal was eliminated when *rad51Δ* was
introduced into our strain. The *RAD51*-dependent DIP results strongly support the formation of
Rad51p-dependent hybrids in *trans*.

We also tested for hybrid formation on the YAC in *trans* by monitoring YAC instability in LW7003.
As expected, no increase in YAC instability was observed in this strain in the absence of galactose
(*Figure 6A*). However, YAC instability increased tenfold upon galactose-induced transcription of
the YAC-GALpr module on chromosome III (*Figure 6A*, black bars). The transcription-induced YAC
instability was dependent on the homology between the YAC and the transcribed YAC sequences
from the YAC-GALpr module on chromosome III, as deletion of the corresponding 1 kb of homology
from the YAC completely suppressed the transcription-induced YAC instability (*Figure 6—figure
supplement 1*). The elevated YAC instability was blocked by RNase H over-expression, indicating the
YAC instability was hybrid dependent (*Figure 6A*, gray bars). YAC instability was also blocked after
introduction of the *rad51Δ* in LW7003 (*Figure 6B*). Thus transcription from the YAC-GALpr module on
chromosome III acted in *trans* to cause the YAC to rearrange through a hybrid- and Rad51p-dependent
mechanism.

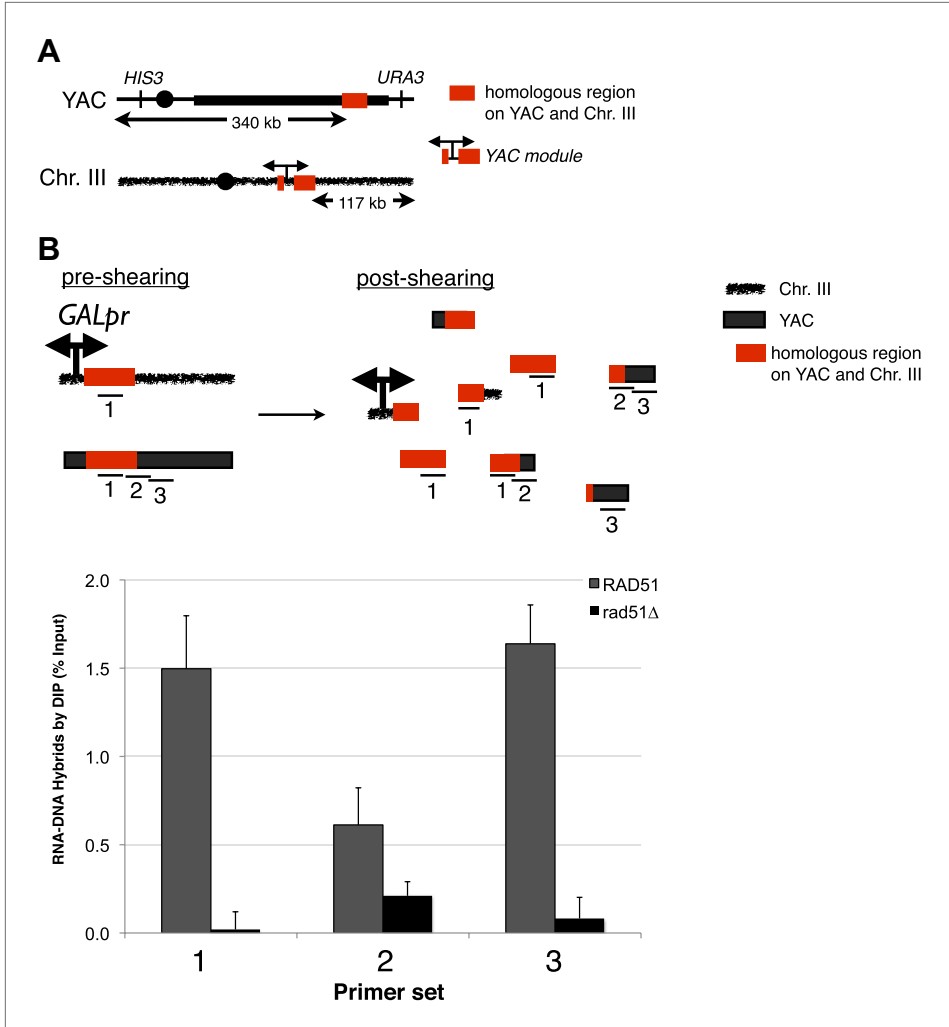

**Figure 5**. Transcription of YAC sequences from chromosome III causes RNA–DNA hybrid formation in *trans* on the YAC. (**A**) Schematic representation of the *trans* assay is depicted. The *GALpr*, selectable marker (CLONAT), and 1.1 kb of yeast artificial chromosome (YAC) DNA was integrated on chromosome III. (**B**) Schematic representation of where the primer sets used to monitor hybrid formation in *trans* are depicted. Hybrid formation is monitored by DIP 2 hr after induction with galactose in *RAD51* and *rad51Δ* strains. Error bars represent standard deviation from two independent experiments.

If the YAC instability induced by the YAC-GALpr module on chromosome III is mediated by hybrids formed in *trans* on the YAC, then these hybrids should lead to a similar distribution of YAC loss and terminal deletion as hybrids induced in *cis*. Indeed hybrids induced in *trans* and in *cis* both lead to a similar distribution of YAC instability events; on average 85% are HIS– URA– (chromosome loss) and 15% are HIS+ URA– (putative terminal deletions). However, the total rate of YAC instability increased only 10-fold by hybrids formed in *trans* (from the YAC-GALpr module on chromosome III) compared to 25-fold by hybrids formed in *cis*. Thus, hybrid formation in *trans* may be less efficient than in *cis*.

While we have assumed that HIS+ URA– clones of LW7003 reflect terminal deletions of the YAC, these clones may have had rearrangements that occurred by an indirect mechanism as a result of hybrid-induced double strand breaks in *cis*. In this model hybrids would form in *cis* at the module on chromosome III and cause DSBs there. These DSBs in *cis* would induce recombination between the YAC sequences on the broken chromosome III and the YAC, resulting in a chromosome III;YAC translocation that has the same genetic phenotype (HIS+ URA–) as YAC terminal deletions (**Figure 6— figure supplement 2**). To determine what fraction of rearrangements may have occurred by this

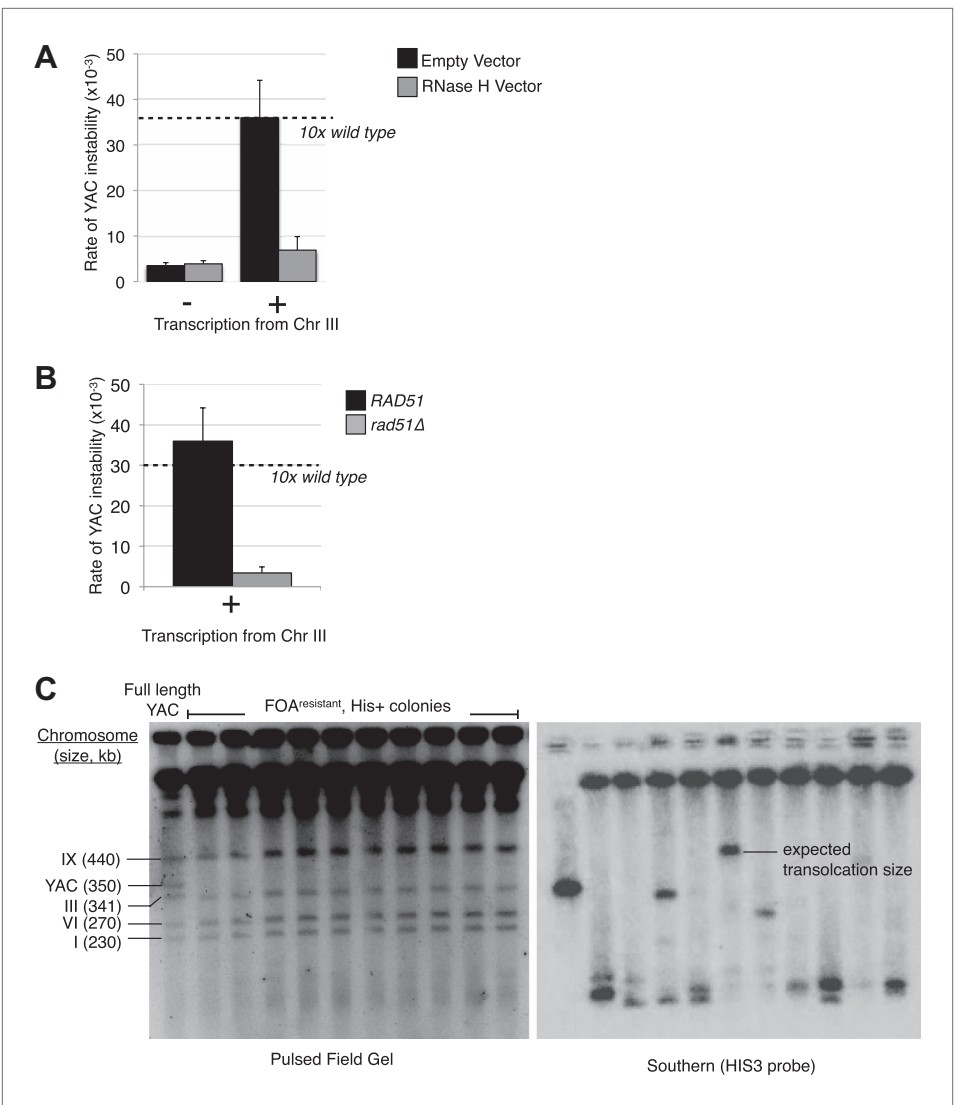

**Figure 6**. Transcription of YAC sequences in *trans* causes hybrid-mediated YAC instability. (**A**) Rates of yeast artificial chromosome (YAC) instability in strains carrying an empty control vector (black bars) or RNase H over-expressing vector (gray bars) showing an increased rate of instability upon induction of transcription that is reduced when RNase H is over-expressed. Error bars represent standard deviation calculated from at least three independent colonies. (**B**) Rate of YAC instability is suppressed when *RAD51* is knocked out. (**C**) Pulse-field gel and Southern analysis with HIS3 probe of FOA^resistant, His+ colonies, showing that 9/10 colonies analyzed have YACs rearranged to a smaller size.

The following figure supplements are available for figure 6:

**Figure supplement 1**. Levels of YAC instability in the *trans* assay with and without a region of homology on the YAC.

**Figure supplement 2**. Schematic representation of an alternative for how HIS+ URA− colonies may arise in the *trans* assay.

indirect mechanism, we performed pulse-field gel and Southern analysis on DNA isolated from 10 independent HIS+ URA− colonies of LW7003. Amongst the 10 YAC rearrangements analyzed, nine were shorter than the existing YAC, consistent with the formation of YAC terminal deletions (*Figure 6C*). Only one rearrangement was the size expected if a chromosome III;YAC translocation had occurred. Thus the structure of most rearranged YACs in LW7003 is consistent with the formation of terminal

 

deletions through the formation of hybrids in *trans*. These results further support our hypothesis that hybrids can form in *trans* by a Rad51p mechanism, causing chromosome instability at sites distinct from the site of hybrid RNA transcription.

## Enhancers and repressors of Rad51p modulate hybrid formation

During homologous recombination, the activity of Rad51p is regulated by a number of factors that modulate Rad51p binding to ssDNA and dsDNA (*Krejci et al., 2003*; *Sugawara et al., 2003*). Because of the importance of such accessory factors for Rad51p function, we wondered whether they might also help regulate Rad51 in hybrid formation. To test this, we deleted positive and negative regulators of Rad51–DNA filament formation.

Rad52p is required for the binding of Rad51p to ssDNA (*Figure 7A*; *Song and Sung, 2000*). Deletion of *RAD52* (*rad52Δ*) in our panel of transcriptional mutants completely suppressed hybrid staining, as assayed by chromosome spreads (*Figure 7B*, *Figure 7—figure supplement 1*). Note that we were unable to test suppression of YAC instability in the double mutants because the *rad52Δ* alone caused substantial hybrid-independent YAC instability, an expected result given its central role in many repair pathways. Nonetheless, the suppression of hybrid staining by *rad52Δ* suggests that hybrid formation is not simply a consequence of rogue activity by Rad51p but rather occurs as part of the canonical Rad51p repair pathway.

A number of inhibitors of Rad51p have been identified. *SRS2* is a helicase involved in removing Rad51p filaments formed on ssDNA (*Krejci et al., 2003*), and Rad54p and Rdh54p are two translocases that promote the removal of Rad51p from double-stranded DNA (*Shah et al., 2010*). We wondered whether these inhibitors might help suppress the rogue hybrid-forming activity of the Rad51p pathway in wild-type cells. To test this we deleted *SRS2, RAD54*, and *RDH54* from cells and measured hybrid formation and YAC instability. Neither single nor double deletions of *RAD54* and *RDH54* significantly increased hybrid formation or YAC instability (*Figure 8A,B*, *Figure 8—figure supplement 1*). In contrast, deletion of *SRS2* increased both hybrid staining and YAC instability. Both of these phenotypes of the *srs2Δ* were suppressed in the *srs2Δ rad51Δ* mutant (*Figure 8C*). Thus, Srs2p antagonizes the hybrid-forming activity of the Rad51p pathway and represents another mechanism by which cells protect their genome against hybrid formation.

The hybrid staining pattern in *srs2Δ* nuclei was reminiscent of the pattern observed in *sin3Δ* cells, exhibiting an apparent enrichment of RNA–DNA hybrids at the RDN locus on chromosome XII, the site of 150 tandem rDNA copies (*Wahba et al., 2011*). We measured rDNA instability by monitoring the rate of excision of a *URA3* marker inserted at the RDN locus (*Heidinger-Pauli et al., 2010*). In *srs2Δ* cells, the rate of rDNA instability is twenty threefold greater compared to wild-type cells, a marked increase in instability as compared to the fourfold increase in YAC instability (*Figure 8C,D*). Together these results suggest that Srs2p has a particularly important role in protecting the highly transcribed rDNA locus against Rad51p-dependent hybrid formation and repeat instability.

## Discussion

In this study we describe a compelling series of observations that demonstrate an in vivo role for Rad51p in promoting formation of RNA–DNA hybrids. First, cytological detection of RNA–DNA hybrids in RNA biogenesis mutants is dramatically suppressed when *RAD51* is deleted. Second, Rad51p is required for the associated hybrid-mediated instability of a YAC in the RNA biogenesis mutants. Third, Rad51p is required for the hybrid formation and YAC instability that results from galactose-induced transcription of specific YAC sequences in our hybrid model locus. Fourth, cofactors canonically known to regulate Rad51p binding and function also modulate hybrid formation. Removal of Rad52p, a positive regulator of Rad51p, blocks hybrid formation. Conversely, the removal of Srs2p, a negative regulator of Rad51 filament formation, increases both hybrid formation and genome instability in a Rad51p-dependent manner. Finally, Rad51p binds to the YAC at the site of hybrid formation proximal to the galactose promoter, in a transcription-dependent manner, prior to any evidence that DSBs have formed. Taken together these results strongly suggest that Rad51p plays a direct role in RNA–DNA hybrid formation.

Establishing a role for Rad51p in the formation of RNA–DNA hybrids marks the first direct in vivo evidence of a factor that facilitates hybrid formation. Indications that a strand-exchange mechanism is important for hybrid formation were suggested by in vitro work on RecA, where RecA catalyzes assimilation of complementary RNA into a homologous region in duplex DNA (*Kasahara et al., 2000*;

 

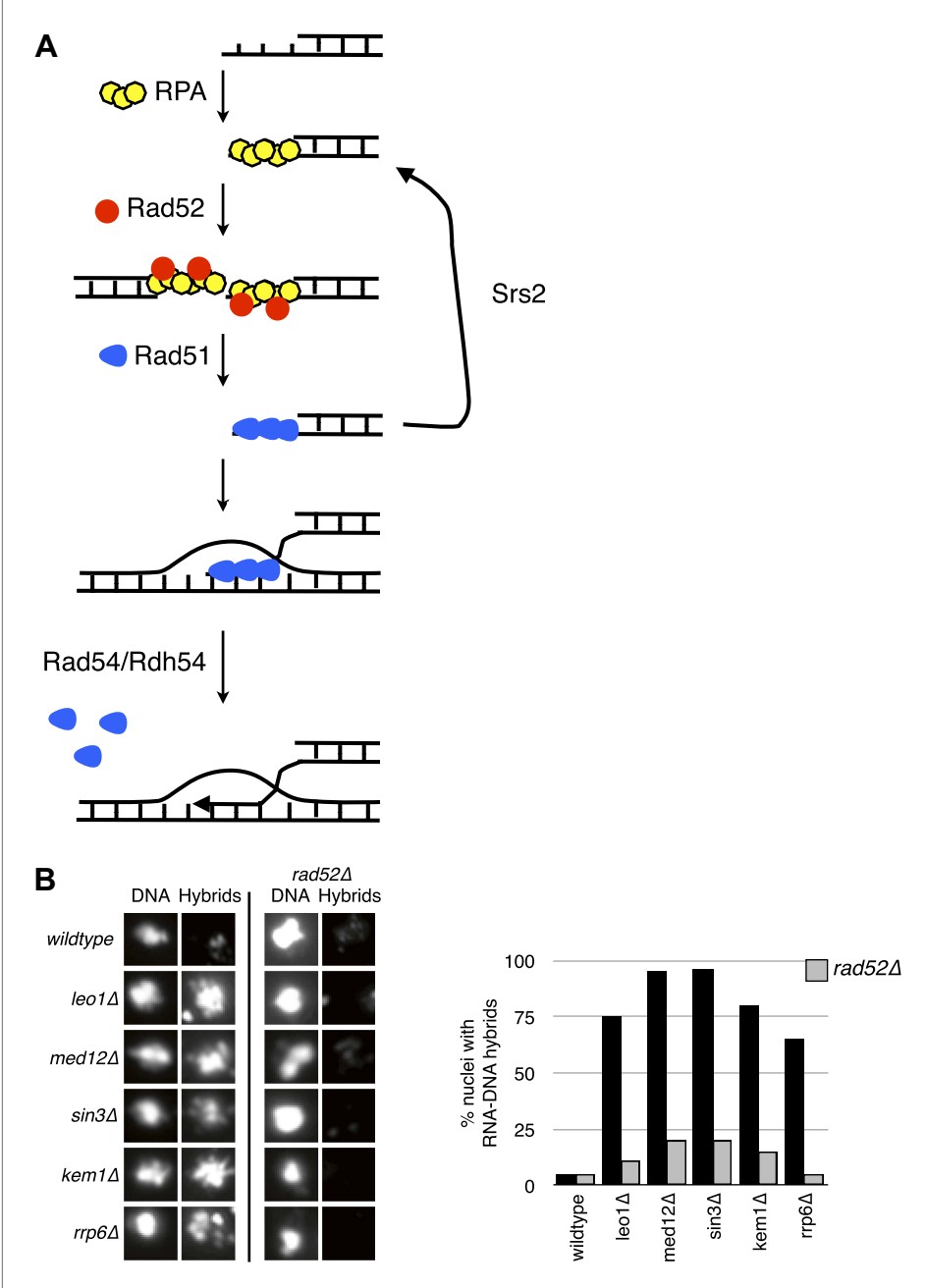

**Figure 7**. Deletion of *RAD52* suppresses RNA–DNA hybrids. (**A**) Schematic showing the major proteins canonically involved in regulating Rad51p binding in DNA repair. Following resection, replication protein A (RPA) polymerizes onto ssDNA. Rad52 then interacts with RPA and catalyzes its exchange for Rad51p. The Rad51–ssDNA filament promotes the pairing and strand exchange reaction with a homologous region in duplex DNA. Srs2p, Rad54p, and Rdh54p all regulate the Rad51 filament by dismantling Rad51 from ssDNA and dsDNA, respectively. (**B**) Representative images of chromatin spreads stained with S9.6 antibody and quantification of nuclei, showing reduced RNA–DNA hybrid staining in mutants with *RAD52* knocked out. A total of 50–100 nuclei from two independent experiments were scored for each genotype.

The following figure supplements are available for figure 7:

**Figure supplement 1**. Larger panels of chromatin spreads showing multiple nuclei of *rad52Δ* mutants stained with S9.6 antibody.

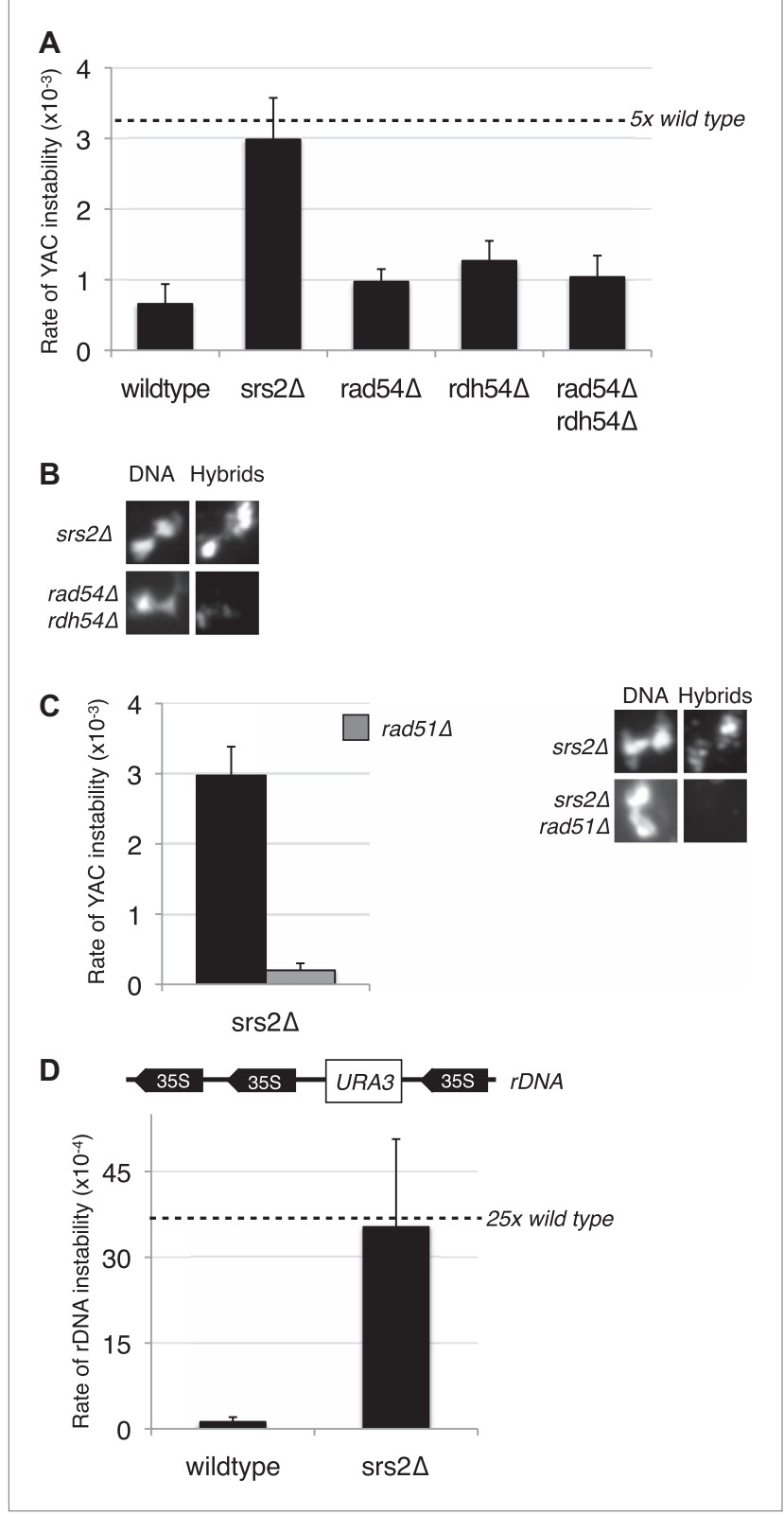

**Figure 8**. Deletion of *SRS2*, but not *RAD54* and *RDH54* increases genome instability and hybrid formation. (**A**) Rate of yeast artificial chromosome (YAC) instability is increased in *srs2Δ* but not in *rad54Δ* and *rdh54Δ* single and double mutants. Error bars represent standard deviation calculated from at least six independent colonies. (**B**) Representative images of chromatin spreads stained with S9.6 antibody, showing increased RNA–DNA hybrid

*Figure 8. Continued on next page*

*Figure 8. Continued*

staining in *srs2Δ*. (**C**) Left panel: Rate of YAC instability in *srs2Δ* is suppressed when *RAD51* is knocked out. Right panel: Hybrid staining is also reduced in the *srs2Δ rad51Δ* double mutant. (**D**) *srs2Δ* mutants with URA3 integrated at the rDNA were assayed for loss of the URA3 marker, showing increased instability. Error bars represent standard deviation calculated from at least six independent colonies.

The following figure supplements are available for figure 8:

**Figure supplement 1**. Larger panels of chromatin spreads showing multiple nuclei of *srs2Δ* and *rdh54Δrad54Δ* mutants stained with S9.6 antibody.

*Zaitsev and Kowalczykowski, 2000*). Interestingly, previous in vivo studies showed that over-expressing Rad51p can compromise genome integrity (*Richardson et al., 2004*; *Shah et al., 2010*). We suggest that this instability may result in part from the ability of Rad51p to promote RNA–DNA hybrid formation. This deleterious activity of Rad51p is also intriguing because a number of studies have shown that *RAD51* expression is up-regulated in tumor cells (*Klein, 2008*). The increased expression is part of a coordinated up-regulation of DNA repair proteins in response to increased damage in cancerous cells (*Zhou and Elledge, 2000*). These high levels of Rad51p have been interpreted as evidence for Rad51p acting as a tumor suppressor by ensuring non-faulty repair of DNA damage. However, our results are consistent with a different interpretation: high levels of Rad51p may be oncogenic, driving hybrid-mediated genomic instability that promotes carcinogenesis.

How can Rad51p promote hybrid formation? In the forward reaction, the conventional mechanism for Rad51p action, it forms a filament on ssDNA, finds a homologous region of dsDNA, and then catalyzes a strand exchange (*Sung, 1994*). By analogy, Rad51p may form a filament on RNA and promote its invasion of dsDNA. However, the bacterial studies of RecA suggest an alternative mechanism in which Rad51 catalyzes RNA–DNA hybrid formation through an inverse strand exchange reaction. In this case, RecA first binds to ssDNA in a gap, forms a filament on adjacent dsDNA, and then promotes pairing and exchange with complementary RNA. Each of these mechanisms has strengths and weaknesses to explain our current findings. For example, the forward reaction but not the inverse strand exchange predicts the association of Rad51p with the hybrid locus should depend upon the induction of the hybrid forming RNA, as we observe. Conversely, the DNA-based inverse strand reaction more easily explains the role of DNA-dependent Rad51p cofactors like Rad52p. Furthermore, in vitro RecA is unable to catalyze the forward reaction with RNA. Clearly an exciting future direction will be to determine the biochemical nature of Rad51p-mediated hybrid formation by either of these mechanisms or an alternative mechanism like stabilization of the R-loop by binding to the RNA–DNA hybrid or the extruded DNA strand (*Figure 9A–C*).

Here we show that Rad51p allows transcripts of YAC sequences generated on chromosome III to act in *trans* to form hybrids and cause hybrid-mediated instability of the YAC. Thus RNA molecules, with the aid of Rad51p can invade duplex DNA to form RNA–DNA hybrids at sites distinct from the site of hybrid RNA transcription. The ability of hybrids to form in *trans* forces a broadening of previous in vivo models for hybrid formation that only considered co-transcriptional mechanisms. The lower level of instability observed in *trans* compared to in *cis* implies that hybrid formation occurs less efficiently in *trans*. A lower efficiency might be expected since the RNA concentration at a *cis* site of hybrid formation will invariably be higher than at a *trans* site of hybrid formation. Indeed, an effect for RNA concentration on hybrid formation has been documented in the in vitro reactions with RecA (*Zaitsev and Kowalczykowski, 2000*). While we show that Rad51p is clearly required to mediate hybrid formation in *trans*, it is likely also important for hybrid formation in *cis*. However, the fact that *rad51Δ* has no effect on hybrid formation in the *rnh1Δrnh201Δ* mutant implies that alternative mechanisms for hybrid formation, presumably in *cis*, exist. Notably, hybrid formation in *trans* might be a more potent promoter of genome instability than hybrid formation in *cis*, particularly for hybrid RNA generated from a highly repetitive element, which in *trans* can cause instability at a plethora of targets.

The formation of hybrids in *trans* has potential applications beyond genome instability. Intriguingly, recent studies in mammalian cells and budding yeast have developed a system for generating targeted DNA breaks using the CRISPR system, where target specificity is determined by a guide RNA complementary to the region of interest (*Jinek et al., 2012*; *DiCarlo et al., 2013*). It is unknown how

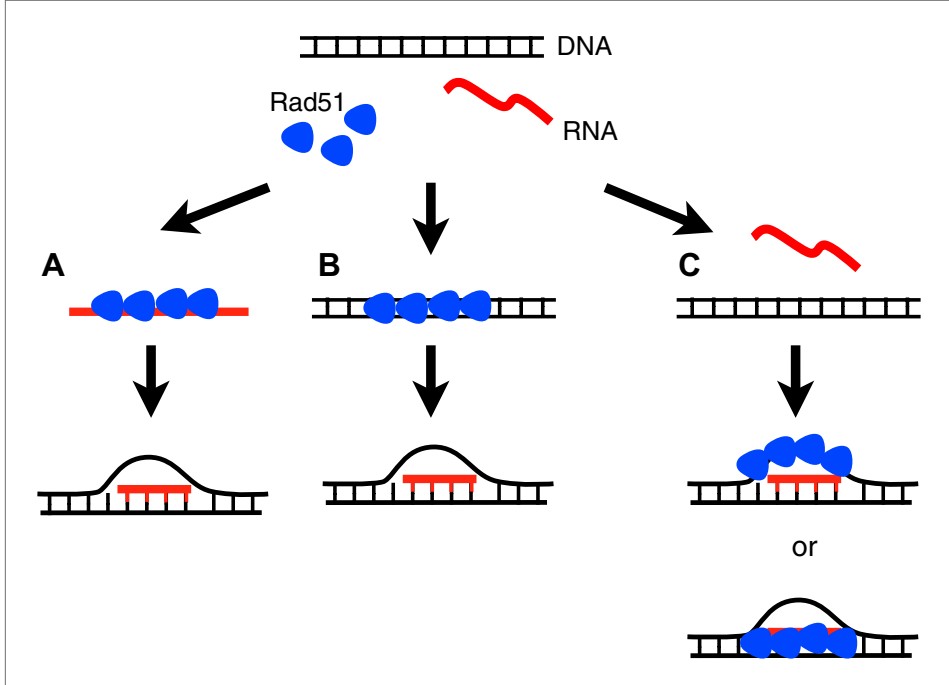

**Figure 9**. Three models for how Rad51p may mediate RNA–DNA hybrid formation. (**A**) In the forward reaction, Rad51p polymerizes onto RNA, and mediates strand exchange with homologous DNA, forming an RNA–DNA hybrid. (**B**) In the inverse reaction, Rad51p forms a filament on dsDNA and promotes strand exchange with homologous RNA. (**C**) A third alternative is that Rad51 forms a filament on the extruded ssDNA, stabilizing an open D-loop that allows RNA to bind to homologous sequences.

the guide RNA finds and hybridizes to its homologous DNA region. From our work, it is clear that these two steps could be mediated through Rad51p. Additionally, positive roles for hybrid formation in modulating transcription state have been found in both human cells and fission yeast (*Ginno et al., 2012*; *Nakama et al., 2012*). In fission yeast, there is evidence that RNA–DNA hybrids may provide a platform for RNAi-mediated heterochromatin formation, driving transcriptional silencing (*Nakama et al., 2012*). If transcripts from one locus can induce hybrid formation and modulate heterochromatin formation at a homologous locus, this could provide a rapid mechanism for the silencing of repetitive elements—including transposable elements—in a genome.

Cells have developed a number of mechanisms to keep RNA–DNA hybrids in check. We provide evidence for a novel Srs2p-dependent pathway that limits the formation of Rad51p-dependent hybrids. We show that deletion of *SRS2* causes Rad51-dependent hybrid formation and YAC instability. Srs2p interacts directly with Rad51p, dismantling inappropriate Rad51p filaments from DNA (*Krejci et al., 2003*; *Burgess et al., 2009*). By removing Rad51p from DNA, Srs2p could potentially inhibit hybrid formation by a mechanism like the inverse strand exchange. Alternatively, Srs2p might be playing a direct role in dismantling RNA–DNA hybrids. Its bacterial homolog UvrD has been shown to catalyze the unwinding of RNA–DNA hybrids in vitro (*Matson, 1989*).

The work reported here, coupled with previous studies, has revealed that cells possess at least four lines of defense against RNA–DNA hybrids: (1) suppressing the deleterious hybrid-forming activity of Rad51p by Srs2p; (2) suppressing the amount of RNA in the nucleus with hybridization potential through proper RNA biogenesis; (3) unwinding RNA–DNA hybrids by helicases such as Sen1p; and (4) degrading RNA in RNA–DNA hybrids by RNases H. The fact that hybrids form when any one of these anti-hybrid pathways is abrogated clearly indicates that they are not completely functionally redundant. The increased propensity for hybrids to form and cause instability at the rDNA locus in *srs2Δ* cells is particularly intriguing, as it may be that the abundance of rRNA, as well as homologous rDNA loci, makes this region a particularly good substrate for Rad51p-mediated hybrid formation. When Rad51p activity is no longer limiting because of inactivation of Srs2p, hybrid

formation may overwhelm the anti-hybrid activities of RNases H or Sen1p. Alternatively, the RNases H or Sen1 may be occluded from the nucleolus, making the rDNA locus more susceptible to hybrid formation by elevated Rad51p activity. It will be exciting to map where hybrids form in RNase H, *sen1-1* and *srs2Δ* mutants to elucidate whether these different anti-hybrid systems are dedicated to protect different regions of the genome.

## Materials and methods

### Yeast strains, media, and reagents

Full genotypes for the strains used in this study are listed in *Supplementary file 1A*. Strain LW6811, the YAC-GALpr strain, was made by integrating the GAL1-10 promoter along with the selectable marker CLONAT at site 323,280 kb on the YAC. The *trans* YAC module in LW7003 encompasses 1 kb of the YAC, along with the *GALpr* and CLONAT marker integrated on chromosome III in place of the *BUD5 (YCR038C)* open reading frame. All integrations were done using standard one-step PCR techniques. The 70mers used for integration are listed in *Supplementary file 1B*. The empty control and RNase H plasmids used are *2µ* plasmids, previously described in *Wahba et al., 2011*. Yeast strains were grown in YEP or minimal media supplemented with 2% glucose. 5-Fluoroorotic (5-FOA) was purchased from BioVectra (Charlottetown, PE).

### Quantitative assay for YAC instability

Cells were dilution streaked out on SC-URA plates to select for the YAC terminal marker (URA3). Single colonies were then picked and resuspended in 0.5 ml of water, diluted, and $10^5$ cells were plated onto 5-FOA and –HIS 5-FOA plates. Plating efficiency was monitored by plating 200 cells onto rich media plates. Plates were incubated at 30°C for 3 d after which the number of colonies formed on each plate was counted. The number of colonies that grow on 5-FOA, normalized for plating efficiencies, is a measure of the rate of events.

### Chromosome spreads and microscopy

Chromosome spreads were performed as previously described (*Wahba et al., 2011*). Slides were incubated with the mouse monoclonal antibody S9.6 directed to RNA–DNA hybrids, and available in the hybridoma cell line HB-8730. The primary antibody was diluted 1:2000 in blocking buffer (5% BSA, 0.2% milk, 1× PBS) for a final concentration 0.25 µg/ml. The secondary Cy3-conjugated goat anti-mouse antibody (No. 115-165-003) was obtained from Jackson ImmunoResearch (West Grove, PA) and diluted 1:2000 in blocking buffer. Indirect immunofluorescence (IF) was observed using an Olympus IX-70 microscope with a 100×/NA 1.4 objective, and Orca II camera (Hamamatsu, Bridgewater, NJ).

### Liquid assay for YAC instability with galactose induction

Cells were picked from SC-URA plates, resuspended in SC-URA media, and grown to saturation. Fresh YEP or -URA media with 2% lactic acid, 3% glycerol was inoculated to an optical density (OD) of ~0.3, and allowed to double to an OD of ~1.0. Galactose was then added to a final concentration of 2%. Cells were shaken at 30°C and then plated onto 5-FOA 0, 2 and 5 hr after induction with galactose. Plating efficiency was monitored by plating 200 cells onto rich media plates.

### Dot blotting with S9.6 antibody

Genomic DNA was isolated using the Qiagen Genomic DNA kit (Qiagen, Hilden, Germany). Roughly 1 µg of DNA was resuspended to a final volume of 50 µl in nuclease-free water, and spotted directly onto a nylon GeneScreen Plus membrane (NEF988; PerkinElmer, Waltham, MA) using a Bio-Dot Microfiltration Apparatus (Bio-Rad, Hercules, CA). The membrane was UV-crosslinked and blocked with 5% 1× PBS/0.1% Tween-20 prior to incubation with primary and secondary antibodies. A 5 µg aliquot of S9.6 antibody was used for the primary, and a 25,000× dilution of goat anti-mouse HRP (Bio-Rad) was used as the secondary. The HRP signal was developed with Clarity Western ECL Substrate (Bio-Rad) and exposed to autoradiography film.

### Quantitative reverse transcriptase PCR

Total RNA was isolated using an RNeasy Mini Kit (Qiagen). Reverse transcriptase was carried out with specified primer pairs using the OneStep RT-PCR Kit (Qiagen) and quantified using SYBR Green (Invitrogen, Carlsbad, CA) and the DNA Engine Opticon Continuous Fluorescence Detection System (CMJ Research).

## DNA immunoprecipitation (DIP)

DIP analysis was performed as previously described (*Mischo et al., 2011*; *Alzu et al., 2012*). Briefly, 150–200 µg of genomic DNA isolated using the Qiagen Genomic DNA kit was sonicated, precipitated, and resuspended in 50 µl of nuclease-free water. Then 350 µl of FA buffer (1% Triton X-100, 0.1% sodium deoxycholate, 0.1% SDS, 50 mM HEPES, 150 mM NaCl, 1 mM EDTA) was added to the DNA, and incubated for 90 min with 5 µg of S9.6 antibody prebound to magnetic protein A beads. Beads were then washed and the DNA eluted according to standard ChIP protocols. % RNA–DNA hybrid amounts were quantified using quantitative PCRs on DNA samples from DIP and total DNA with the DyNAmo HS SYBR Green qPCR kit (Thermo Scientific, Waltham, MA).

## Chromatin immunoprecipitation (ChIP)

Cells used for ChIP experiments were grown in YEP media with 2% lactic acid, 3% glycerol and collected either before galactose was added (−Gal) or 2 hr after the addition of galactose at a final 2% concentration (+Gal). Standard ChIP was performed as described previously (*Unal et al., 2004*). Briefly, $5 \times 10^8$ cells were crosslinked in 1% formaldehyde for 30 min at room temperature. Chromatin was sheared 20 times for 45 s each (settings at duty cycle: 20%, intensity: 10, cycles/burst: 200) with 30 s of rest in between using a Covaris S2. Immunoprecipitation of Rad51-HA or untagged Rad51p was done with anti-HA antibody (Roche, Mannheim, Germany) or anti-Rad51p polyclonal antibody (Santa Cruz, Dallas, TX). Immunoprecipitation of γ-H2a.X was done with anti-γ-H2AX (Abcam, Cambridge, UK). A no primary antibody control is also run to ensure specificity. Appropriate dilutions of input and immunoprecipitated DNA samples were used for PCR analysis to ensure linearity of the PCR signal. PCR and data analysis was carried out as described previously (*Unal et al., 2004*). With the exception of the experiment shown in *Figure 3—figure supplement 1* which was carried out once, all experiments were done at least twice and a representative data set is shown. ChIP primers are listed in *Supplementary file 1B*.

## Pulse-field gel electrophoresis and Southern analysis

Yeast genomic DNA was prepared in 1% pulse-field grade agarose plugs (SeaPlaque 50100) and resolved as previously described (*Schwartz and Cantor, 1984*) with a Bio-Rad CHEF-DR III system. The following parameters were used: 6 V/cm, 120° angle, 20–50 s switch times, 17 hr at 14°C. For Southern analysis, gels were transferred onto a GeneScreen Plus membrane (PerkinElmer NEF988) and probed with a 0.5 kb fragment containing *HIS3* sequence.

## Quantitative assay for rDNA instability

Cells were dilution streaked out on SC-URA. The rate of rDNA instability was calculated from 5-FOA plates as described above for YAC instability.

# Acknowledgements

We would like to thank Chen-Ming Fan, Vincent Guacci, Anjali Zimmer, Gamze Camdere, and Jeremy Amon for constructive comments on the manuscript. We thank members of the Koshland Lab for technical assistance and helpful discussions.

# Additional information

### Funding

| Funder | Author |
| --- | --- |
| Howard Hughes Medical Institute | Steven K Gore |

The funders had no role in study design, data collection and interpretation, or the decision to submit the work for publication.

### Author contributions

LW, Conception and design, Acquisition of data, Analysis and interpretation of data, Drafting or revising the article; SKG, Conception and design, Acquisition of data, Analysis and interpretation of data; DK, Conception and design, Drafting or revising the article

## Additional files

**Supplementary files**
• Supplementary file 1. (**A**) Description of strains used in this study. (**B**) Chromatin immunoprecipitation (ChIP)/DNA immunoprecipitation (DIP) primers and integration primers for integration of the GALpr in LW6811.

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
