## [Decision Letter]

Thank you for choosing to send your work entitled “The homologous recombination machinery modulates the formation of RNA-DNA hybrids and associated chromosome instability” for consideration at *eLife*. Your article has been favorably evaluated by a Senior editor and 3 reviewers, one of whom is a member of our Board of Reviewing Editors.

The Reviewing editor and the other reviewers discussed their comments before we reached this decision, and the Reviewing editor has assembled the following comments to help you prepare a revised submission.

This study describes the role played by *S. cerevisiae* Rad51 in promoting the formation of RNA:DNA hybrid structure, especially in mutant backgrounds that are defective in different stages of RNA biogenesis. Thus while in wt cells loss of Rad51 has no effect on hybrid formation (visualised by immunofluorescence using RNA:DNA hybrid specific antibody), in various mutant cells loss of Rad51 causes loss of hybrid signal (Figure 1). This data is then complemented by use of an integrated human YAC from which recombination rates can be measured (based on a previous paper from Wahba et al, Mol Cell 2011).

The key experiments are an extension of the YAC assay to include an inducible GAL1 promoter in the YAC. Induction by galactose growth promotes bidirectional transcription that correlates with YAC instability, an effect lost by deletion of RAD51 or RNase H over expression. Rad51 is shown to specifically localize to the YAC GAL1 promoter region (Figures 2 and 3). A case is made that this localization is hybrid dependent but not caused by recruitment to DNA damage sites (Figure 4). Another important result (Figure 5) is that these hybrid-induced DNA damage effects can be caused *in trans* by placing the GAL1 promoter plus adjacent human sequence on a yeast chromosome and then showing instability of the YAC (lacking GAL1 promoter). Finally a gene known to antagonise Rad51, Srs2, is also shown to cause hybrid formation in wt cells when deleted, presumably because Rad51 is more active.

Overall these results are of great interest as they point to the role of factors (Rad51/52) in enhancing hybrid formation rather than operating to restrict hybrids (e.g., RNase H and Sen1). The trans effects are also important as a mechanism is implied for how RNA might target duplex DNA to form hybrids. This could be relevant to gene silencing by RNAi pathways in higher eukaryotes.

The major issue with these studies is proving that Rad51 directly promotes R-loops or instead is recruited to R-loop induced sites of DNA damage to promote recombination. The referees feel that the case for a direct effect needs tightening by additional experiments as listed below.

1) Statistical analysis of how many nuclei over a field of cells show positive immunofluorescence for the S9.6 antibody is required whenever this analysis is employed (e.g., Figures 1, 4 and 6, and 7B&C).

2) Controls for the specificity of the S9.6 signal. Is it RNase H sensitive? Is it transcription dependent? Is it nuclear specific?

3) Better analysis of GAL induced ncRNA over the YAC human sequence. Showing fold change post GAL induction is too imprecise. Ideally RNAseq or strand specific Northern analysis is needed.

4) Direct detection of R-loops over the ncRNA positions (as in 2) should be obtained by using S9.6 in chromatin or genomic DNA immunoprecipitation experiments (ChIP or DIP).

This is especially important for the trans experiments (Figure 5).

5) Controls for how much (if any) homologous sequence are required for GAL induction of ncRNA to work *in trans* to promote YAC instability.

6) Test SEN1 and TOP1/2 mutation on RAD51 dependent R-looping.

Finally we feel that it is impossible to exclude the possibility that loss of RAD51 has a combined effect on both recombination and the promotion of RNA:DNA hybrids. The paper’s claims should acknowledge this possibility. Indeed one neat way to address this issue might be to incorporate an HO cleavage site near to the GAL human DNA insert in Chr III. If homologous recombination at the target YAC region is promoted by a combination of HO endo and GAL transcription then this would suggest that HR employs R-loops to promote recombination. If such an experiment is doable this might suggest that HR involves R-loop formation. Such an experimental approach would be very helpful to the paper’s final impact in the field.

---

## [Author Response]

*1) Statistical analysis of how many nuclei over a field of cells show positive immunofluorescence for the S9.6 antibody is required whenever this analysis is employed (e.g., Figures 1, 4 and 6, and 7B&C)*.

We've quantified the number of nuclei that stain positively and represent it as the percent of the total number of nuclei scored from two independent experiments.

*2) Controls for the specificity of the S9.6 signal. Is it RNase H sensitive? Is it transcription dependent? Is it nuclear specific*?

We agree that the specificity of the S9.6 signal was not outlined clearly in the paper and we have changed the text to make this point clearer. The specificity of the antibody to RNA-DNA hybrids in chromatin spreads was demonstrated in our first paper (33) by showing that both in vivo over-expression as well as treatment of slides with RNase H reduces the signal to near background levels. Note that in the chromatin spreads, all cellular components with the exception of the chromatin is removed, and so the signal observed in the spreads are nuclear-specific. Furthermore, for a representative mutant we corroborated our results using the antibody to probe total nucleic acid on a membrane. Additionally, sensitivity of the S9.6 signal to RNase H treatment in ChIP/DIP was previously established by other labs (Hage et al., 2010 and [19]). Given the specificity of the antibody for RNA-DNA hybrids in our assays, the signal must be transcription dependent; otherwise, how does one make the RNA component of the hybrid? Furthermore, using our model locus assay, we also show transcription dependence of the elevated S9.6 signal as it occurs only upon galactose induction of transcription in the region.

*3) Better analysis of GAL induced ncRNA over the YAC human sequence. Showing fold change post GAL induction is too imprecise. Ideally RNAseq or strand specific Northern analysis is needed*.

While further analysis of the transcripts is interesting, we believe that the level of analysis presented in the paper is sufficient for the question we are addressing: i.e., is inducing high levels of transcription on the YAC sufficient for promoting hybrid formation and subsequent instability? The purpose of the RNA analysis is to define the domain of transcription that serves as a basis to evaluate the DIP and Rad51p ChIP. Importantly, the RNA analysis predicts hybrid formation and Rad51p binding on both sides of the GALpr, as we observe. Hybrids are confined to the region of transcription as expected, while interestingly Rad51p extends beyond the transcript/hybrid domain. The latter is not unprecedented as both in vitro and in vivo results indicate that Rad51p can polymerize into double stranded DNA from gaps, supporting the possibility that Rad51p during the formation or stabilization of R loops polymerizes onto adjacent DNA.

*4) Direct detection of R-loops over the ncRNA positions (as in 2) should be obtained by using S9.6 in chromatin or genomic DNA immunoprecipitation experiments (ChIP or DIP)*.

*This is especially important for the trans experiments (Figure 5)*.

As discussed above, we now directly detect RNA-DNA hybrids on the YAC in the model locus, both *in cis* as well as *in trans* using DIP. We’ve demonstrated that these hybrids are both transcription dependent and are dependent on Rad51p. These data have been added to figures where appropriate.

*5) Controls for how much (if any) homologous sequence are required for GAL induction of ncRNA to work* in trans *to promote YAC instability*.

We've included the result in which complete deletion of the homologous sequence on the YAC suppresses the YAC instability associated with induction of YAC-GALpr module on chromosome III, validating that the homology is required to promote YAC instability.

*6) Test SEN1 and TOP1/2 mutation on RAD51 dependent R-looping*.

The dependence of these mutants on Rad51 is definitely interesting; however, we believe it is not needed for this paper.

*Finally we feel that it is impossible to exclude the possibility that loss of RAD51 has a combined effect on both recombination and the promotion of RNA:DNA hybrids. The paper's claims should acknowledge this possibility. Indeed one neat way to address this issue might be to incorporate an HO cleavage site near to the GAL human DNA insert in Chr III. If homologous recombination at the target YAC region is promoted by a combination of HO endo and GAL transcription then this would suggest that HR employs R-loops to promote recombination. If such an experiment is doable this might suggest that HR involves R-loop formation. Such an experimental approach would be very helpful to the paper's final impact in the field*.

We don't disagree with the possibility that RNA-DNA hybrids may promote recombination in addition to hybrid formation. Unlike hybrid formation where we provide direct evidence, our experiments do not provide any evidence in support of or against a role for hybrids in recombination. Additionally, there is reason to believe that hybrids do not play a role in recombination, as you would expect that over-expression of RNase H in mutant cells (such as repair or checkpoint defective mutants) should inhibit repair and would lead to an even higher level of instability (in particular chromosome loss), which we do not find to be the case (33).

What our data does indicate is that RAD51 is involved in hybrid formation and that this Rad51-mediated hybrid formation can occur *in trans*. Our findings provide the first *in vivo* evidence of a positive factor promoting hybrid formation, and the finding that hybrids can also occur *in trans* has far-reaching implications for understanding how hybrids form and what other cellular mechanisms their formation may be modulating. We strongly believe that either of these findings is a significant contribution, and together make this manuscript a critical advance in the field.